# 3D-BCLAM: A Lightweight Neurodynamic Model for Assessing Student Learning Effectiveness

**DOI:** 10.3390/s24237856

**Published:** 2024-12-09

**Authors:** Wei Zhuang, Yunhong Zhang, Yuan Wang, Kaiyang He

**Affiliations:** 1School of Computer Science, Nanjing University of Information Science and Technology, Nanjing 210044, China; zhangyh@nuist.edu.cn; 2School of Teacher and Education, Nanjing University of Information Science and Technology, Nanjing 210044, China; momo@nuist.edu.cn; 3School of Mathematics and Physics, Xi’an Jiaotong-Liverpool University, Suzhou 215123, China; Kaiyang.He22@student.xjtlu.edu.cn

**Keywords:** emotion recognition, brain-computer interface, student learning effectiveness

## Abstract

Evaluating students’ learning effectiveness is of great importance for gaining a deeper understanding of the learning process, accurately diagnosing learning barriers, and developing effective teaching strategies. Emotion, as a key factor influencing learning outcomes, provides a novel perspective for identifying cognitive states and emotional experiences. However, traditional evaluation methods suffer from one sidedness in feature extraction and high complexity in model construction, often making it difficult to fully explore the deep value of emotional data. To address this challenge, we have innovatively proposed a lightweight neurodynamic model: 3D-BCLAM. This model cleverly integrates Bidirectional Convolutional Long Short-Term Memory (BCL) and dynamic attention mechanism, in order to efficiently capture emotional dynamic changes in time series with extremely low computational cost. 3D-BCLAM can achieve a comprehensive evaluation of students’ learning outcomes, covering not only the cognitive level but also delving into the emotional dimension for detailed analysis. Under testing on public datasets, 3D-BCLAM has demonstrated outstanding performance, significantly outperforming traditional machine learning and deep learning models based on Convolutional Neural Networks (CNN) and Recurrent Neural Networks (RNN). This achievement not only validates the effectiveness of the 3D-BCLAM model, but also provides strong support for promoting the innovation of student learning effectiveness assessment.

## 1. Introduction

Emotions are the psychological reactions that humans experience when faced with external events, such as happiness or sadness. It has a significant impact on our perception, memory, and expression. Nowadays, emotion recognition technology is becoming increasingly important in fields such as human-computer interaction and emotion computing. Whether it’s smart speakers, mobile phones, or our commonly used social media, they are all using this technology to analyze users’ emotions. Based on the results of emotion recognition, these systems can better understand our needs and provide us with more thoughtful and personalized services.

During the learning process, students’ emotions play a crucial role. Research has found that when students are in a good mood, such as feeling interested or enjoying learning, they are more proactive, engaged, and learn better. On the contrary, if students feel anxious or depressed, their learning will be hindered and their grades may also decline. Therefore, if we can take into account students’ emotions when evaluating their learning effectiveness, we can have a more comprehensive understanding of their learning status. At the same time, teachers can develop more suitable teaching methods based on each student’s emotional and cognitive needs. This assessment method that combines emotions and learning can assist teachers in teaching.

Currently, facial expressions are a relatively common method for human emotion recognition. This approach is based on computer vision and uses methods such as machine learning for classification and prediction. Avula et al. [1] suggested a methodology where facial emotions are initially learned through a CNN. Subsequently, an emotion-to-speech model is trained. Ultimately, this approach integrates hand gestures with the recognized facial emotions to comprehend and produce emotion alongside speech. Gupta et al. [2] focused on facial emotion recognition using thermal imaging. The primary goal of this work is to enhance the efficiency, robustness, and accuracy of emotion recognition systems to improve human-computer interaction, mental health monitoring, personalized education, marketing effectiveness, and security. The proposed CNN architecture is designed to recognize human emotions by classifying facial images into six distinct categories (happiness, sadness, anger, fear, disgust, and surprise). Foo et al. [3] devised a system for recognizing facial emotions aimed at detecting mental stress. Salma et al. [4] presented a machine learning model based on CNN to predict the face expressions. However, this method of recognizing human emotions through facial expressions has certain disadvantages. This is because facial expressions are not necessarily the most intuitive expression of human emotions. Some people may express emotions that are different from their inner thoughts under special circumstances. Therefore, physiological signals, as one of the characteristics of the human body, can intuitively display the inner activities of the human body and are an important basis for emotion detection.

The various life activities of humans rely on the core central control system: the brain. The human brain is composed of tens of thousands of neurons, and the activity between neurons generates electrical signals. In 1924, Berger measured the first human electroencephalogram (EEG) through the activity of electric eels. This discovery greatly promoted human interest in the study of the connection between the human body and the outside world, giving rise to brain computer interface technology.

In recent years, emotion detection research using EEG signals has gained significant attention. Analyzing EEG signals allows for the direct observation and quantification of attention levels, offering the advantages of objectivity and accuracy. Some early proposed methods took advantage of the huge advantages of EEG signals in feature band matching, such as Alpha, Beta, Theta, Gamma, etc. They all improved traditional frequency band power feature extraction algorithms by introducing machine learning methods, optimizing the algorithm’s processing and results through classic frameworks such as K-Nearest-Neighbor (KNN), K-means, Support Vector Machines (SVM), and auto regressive models. Using artificial neural networks to automatically capture the time-domain and frequency-domain features of EEG signals, and effectively extract and analyze these features, thereby directly learning the complex nonlinear relationship between the original EEG signal and physiological features; Meanwhile, these technologies can be trained through a large amount of data, thus having high prediction accuracy. Li et al. [5] used Fast Fourier Transform (FFT) and Continuous Wavelet Transform (CWT) to extract the features of EEG signals. Liu et al. [6] introduced a novel approach employing a three-dimensional convolutional attention neural network (3DCANN) specifically designed for the emotion recognition based on EEG signals. Zhao et al. [7] proposed Differential Entropy-Recurrent Neural Network- Convolutional Neural Network (DE-CNN-RNN) model to explore emotion recognition.

Graph Neural Network (GNN) is a kind of deep learning models designed to process graph-structured data, which have garnered increasing attention in EEG-based emotion recognition. GNNs excel at capturing the spatial relationships among electrodes in EEG signals, thereby extracting more discriminative features. Guo et al. [8] proposed a novel model that integrates graph neural network-based prototype representation across multiple source domains with clustering similarity loss. Zhong et al. [9] proposed a regularized graph neural network (RGNN) for EEG-based emotion recognition, which considerd the biological topology among different brain regions to capture both local and global relations among different EEG channels. On the other hand, the Transformer architecture, renowned for its powerful sequence modeling capabilities, initially achieved remarkable success in the field of natural language processing. More recently, research has begun to utilize a unique self-attention mechanism to efficiently process long-range dependencies and temporal features in EEG signals. These studies typically serialize EEG signals into a sequence of time steps or feature vectors, subsequently leveraging Transformers for feature extraction and classification. Liu et al. [10] designed a hybrid architecture combining CNN and transformer for analyzing human psychological states. However, these methods have certain shortcomings in feature selection and model construction, and one-dimensional features have accuracy defects in time, frequency band and spatial feature analysis. 

Traditional convolutional neural networks cannot capture precise features in time. At the same time, in order to cope with major emotional changes in the human body, it is necessary to preprocess the EEG signals. Therefore, we propose a lightweight neurodynamic emotion recognition model: 3D-BCLAM. For the raw dataset, we propose a new pre-processing process and selects differential-entropy as the input feature. The model is based on the CNN and Bi-LSTM module, and combines the dynamic Attention mechanism to classify the features of the emotional dataset. This model has certain advantages in parameter quantity and running time. The model is stable and robust and has good applicability. The primary contributions of this study are outlined below:This paper proposes a novel preprocessing process for EEG emotional raw data, which involves designing a bandpass filter to optimize signal quality and stacking the processed data into a three-dimensional form to enhance the model’s ability to represent features.This paper introduces the Bi-LSTM model, which can simultaneously consider the forward and backward information of time series data, thereby capturing emotional features in EEG signals more comprehensively.This paper integrates dynamic Attention Mechanism that enables the model to focus more on EEG signal features that are more important for emotion recognition tasks, ignoring irrelevant or redundant information, thereby improving the model’s discriminative power and robustness.Experiment on open-source datasets and real-time monitor: In this study, the robustness of emotion recognition is ensured through rigorous experimental validation using extensive public datasets. Additionally, real-time monitoring is implemented to assess performance across a wide range of samples and scenarios.This paper proposes an evaluation model for student learning effectiveness that integrates emotion classification. The model categorizes students’ various learning states based on calculated scores, and its efficacy has been validated in real-world scenarios.

## 2. Related Work

EEG signals are potential changes recorded from the scalp of humans or other animals, mainly reflecting the electrical activity characteristics of the brain. It is the overall reflection of the electrophysiological activity of brain nerve cells in the cerebral cortex or scalp surface [11]. In the human brain, the cerebral cortex is related to the cognitive level and emotions of the human body, and the analysis of EEG signals is mainly based on the cerebral cortex. When the human brain is active, a large number of neurons in the brain synchronize, and their postsynaptic potentials can be obtained by high-precision EEG sensors, which are local EEG signals. EEG signals are weak bioelectrical signals, with amplitudes mainly ranging from 0 to 60 μV, the frequency distribution is between 0 and 100 Hz. This type of signal has the characteristics of non-stationary and strong randomness. EEG signals from different frequency bands have different characteristics. Table 1 introduces some common bands that can be used for research on emotion recognition.

Regarding the task of recognizing emotions through EEG, current methods can generally be divided into conventional machine learning techniques and deep learning strategies. The traditional EEG-based emotion recognition methods using machine learning typically follow a sequence of steps: data acquisition, preprocessing, feature extraction, feature selection (optional), classifier design and training, and testing and evaluation. Among these steps, feature extraction and classifier design are the core components. Feature extraction aims to distill crucial information related to emotional states from raw EEG signals, while the classifier identifies different emotional states based on these extracted features. Li et al. [12] segmented EEG signals into four frequency ranges utilizing the discrete wavelet transform method, and computed entropy and energy as characteristics for the K-nearest neighbor classifier. Chen et al. [13] introduced an EEG-based emotion identification approach that relies on the Library for Support Vector Machines (LIBSVM) as the classification tool, and the average sentiment recognition rates on DEAP were 74.88% and 82.63%. Zheng et al. [14] transformed the initial one-dimensional EEG vector signal into a two-dimensional matrix signal that incorporated channel position data. Subsequently, they introduced a recognition technique grounded on an Adaptive Neural Decision Tree (ANT). Xiao et al. [15] introduced the Attention-based Temporal Learner with Dynamic Graph Neural Network (AT-DGNN), the results achieved accuracy of 83.74% in arousal recognition and 86.01% in valence recognition.

As mentioned before, traditional machine learning methods rely on manually designed features, which may not fully capture the complex emotional information. The feature extraction process often depends on the researchers’ experience, which may potentially lead to information loss or redundancy. In contrast, 3D-BCLAM eliminates the need for tedious manual feature engineering through integrating Bi-LSTM and dynamic attention mechanisms. The model is capable of automatically learning features from raw EEG signals. Furthermore, traditional machine learning models may exhibit limited generalization capabilities when dealing with complex emotional states and cross-dataset tasks. This is primarily due to overfitting to specific datasets or sensitivity to variations in data distribution. 3D-BCLAM may possess better generalization abilities to adapt to different datasets and task requirements.

## 3. Methods

### 3.1. Baseline Drift Elimination

EEG baseline drift refers to a slow and sustained change in the baseline of a signal (i.e., the potential level when there is no electrical activity) [16]. This change may manifest as an increase or decrease in baseline level, causing the background potential of EEG signals to deviate from the normal zero potential level. When collecting emotional data from subjects, baseline drift can occur due to changes in their physiological state, including breathing, heartbeat, and muscle activity. Baseline drift presents a challenge to the analysis and interpretation of EEG signals, as it may mask real brain electrical activity and diminish the accuracy and reliability of the signals.

Wavelet transform [17,18], as a time-frequency analysis method, is capable of analyzing local characteristics of signals at different scales. To deal with baseline drift in EEG signals, this method can effectively decompose the signal into components at multiple scales, identify and remove the low-frequency components associated with baseline drift, and subsequently reconstruct the signal. This process ensures that useful information in the signal is well-preserved while effectively eliminating baseline drift. The formula of using wavelet transform to remove baseline drift is shown in (1):(1)$$WT\left(a,\tau \right)=\frac{1}{\sqrt{a}}\int_{-\infty }^{\infty }f\left(t\right)*\varphi \left(\frac{t-\tau }{a}\right)dt$$

The variables in wavelet transform contain displacement *τ* and scale *a*, and the scale control signal waveform shrinks while the displacement control signal waveform shifts. Wavelet basis functions constitute a specific set of functions that efficiently decompose signals into their constituent components across various frequencies and time scales. Owing to their unique localization properties, these functions exhibit finite characteristics in both the time and frequency domains, enabling them to precisely capture the local features of signals. When multiplying the wavelet basis function with the signal function, the relationship between the current scale contained in the signal and the corresponding frequency components can be obtained. 

When selecting wavelet basis functions, it is imperative to consider factors such as signal characteristics, orthogonality, compact support, and vanishing moments. Therefore, in this paper, the biore wavelet is used as the wavelet basis function, which is commonly represented as *biore x.y* (where *x* is the reconstruction coefficient and *y* is the decomposition coefficient). Symmetric wavelets can be used for signal decomposition and reconstruction, respectively. The individual wavelets have tight support, and the support length for reconstruction and decomposition is 2*N* + 1 (where *N* is the coefficient *x*, *y*). Here, the reconstruction coefficient is selected as 4 and the decomposition coefficient is selected as 3. The wavelet coefficients after thresholding are used for signal reconstruction. The reconstruction process is the inverse of wavelet decomposition, which combines the processed wavelet coefficients into an EEG signal with baseline drift removed through layer-by-layer upsampling and filtering operations. The reconstructed signal retains as much useful information from the original signal as possible, while eliminating baseline drift and noise. Figure 1 shows the comparison after baseline drift elimination.

As is shown in Figure 1, zero drift caused by both the device itself and external factors interferes with the true EEG signals, making it challenging to accurately capture and analyze brain electrical activity. However, the reconstructed signal, obtained through wavelet transform, eliminates these negative effects, thereby facilitating further processing.

### 3.2. Pre-Processing

The raw signal collected through a portable headset contains a large amount of noise and artifacts, and certain pre-processing is required for the original signal to facilitate subsequent feature extraction and model classification.

The main frequency of EEG is from 0 to 100 Hz [19], and in practical research, different rhythmic bands represent different physiological meanings. In order to obtain the band we want to study, it is necessary to filter the original signal. According to the characteristic table of EEG signals, band-pass filters are mainly used in this paper. A zero-phase band-pass filter based on Bessel filter has been designed to filter out signals of 0.5 to 40 Hz. The filter has an almost constant group delay throughout the entire pass-band, thereby maintaining the waveform of the filtered signal in the pass-band. The Bessel transfer function [20] in the filter aims to obtain a linear phase (i.e., the smoothest delay), and the impulse response has no oscillation characteristics. The Bessel transfer function is implemented by (2):(2)$$T_{n}\left(s\right)=\frac{B_{n}(0)}{B_{n}(s)}$$

Among them, $B_{n}$ is the Bessel polynomial, as (3) shows:(3)$$B_{n}\left(s\right)=\sum_{k=0}^{n}a_{k}s^{k},\;a_{n}=1$$

The expression of the coefficient is shown in (4):(4)$$a_{k}=\frac{\left(2n-k\right)!}{k!\left(n-k\right)!2^{n-k}}$$

n is the order of the Bessel filter, which is used to observe the signal characteristics of EEG signals with non-stationary digital signal processing type using a 4th order Bessel filter.

Due to the possibility of the same frequency domain components of the signal in different time domains, windowing analysis based on Hamming window [21] is required for the signal, which can be achieved through the (5):(5)$$\omega \left(n\right)=\left\{\begin{array}{c} 0.5[1-\cos\left(\frac{2\pi n}{M-1}\right)],0\leq n\leq M-1\\ 0,others \end{array}\right.$$

In this paper, the application of the Hamming window plays a significant role. The Hamming window effectively smooths the EEG signals, significantly reducing spectral leakage and edge effects, thereby enhancing the accuracy of spectral analysis. Furthermore, by applying windowing processing, critical features in the signals are accentuated [22,23], effectively suppressing noise and unnecessary frequency components, which establishes a strong groundwork for the ensuing feature extraction and classification endeavors.

### 3.3. Feature Extraction

To analyze EEG signals in the time domain is the most direct and effective method, which facilitates the extraction of intuitive features. Time domain analysis is mainly based on the time series analysis of EEG signals. 

The differential entropy [24] of sub frequency bands is shown in (6), which describes the temporal variation patterns of different frequency components in EEG signals, which can effectively transform high-dimensional and low signal-to-noise ratio temporal signals (one-dimensional temporal vectors) into descriptive values of several sub frequency bands (scalar values), and has strong discriminative power for emotional and other brain activities.
(6)$$DE=\frac{1}{2}\log(2\pi e\sigma^{2})$$

In Table 1, it is noted that EEG signals within distinct frequency bands reflect varying characteristics of brain activity. Consequently, we performed a bandpass filtering process on the raw EEG signals, utilizing the bandpass filters designed previously, to isolate the signals into their constituent frequency components. The purpose of this breakdown was to reveal the unique brain activity patterns present within various frequency ranges. In particular, we divided the EEG signals into four main categories: theta (4–8 Hz), alpha (8–14 Hz), beta (14–31 Hz), and gamma (31–45 Hz). The frequency specific analysis allows us to gain a deeper understanding of the neural activity behind various cognitive and emotional states [25].

### 3.4. 3D-BCLAM Framework

As is shown in Figure 2, 3D-BCLAM framework includes two primary components: the neurodynamic data generation module and the BCLAM framework. The neurodynamic data generation module is designed for transforming EEG signals into a three-dimensional neurodynamic data format, which is then sequentially sent into the nodes of the BCLAM framework. This facilitates temporal interactions among nodes and enables the learning of temporal features in the data.

The BCLAM framework consists of three layers: two-dimensional convolutional (2D-Conv) layer, Bi-LSTM layer, and dynamic Attention Mechanism layer. This design enables the framework to efficiently capture complex features in both space and time from converted EEG data. The 2D-Conv layer is used for extracting local features from the original 3D data at beginning; Subsequently, the Bi-LSTM layer further processes these features to capture their long-term dependencies over time; After that, the dynamic attention mechanism layer analyzes the correlation between emotional data by focusing on key features.

The output of BCLAM framework is passed through fully-connected layer, which performs the classification of EEG emotional data. The approach introduces a novel and effective solution for emotion recognition from EEG signals by integrating neuromorphic data generation and deep learning methods.

#### 3.4.1. 3D Data Generation

In this paper, we design a specific data transformation way that efficiently converts one-dimensional EEG data into a three-dimensional format. We employ a custom data reshaping algorithm to transform each processed one-dimensional data segment, corresponding to features extracted from distinct frequency bands, into a two-dimensional matrix, denoted as data_2D. This step relies on a particular mapping rule that allocates consecutive elements from the one-dimensional data to specific locations within the two-dimensional matrix, resulting in a spatially structured feature representation. 

To construct a three-dimensional dataset, we stack all these two-dimensional matrices along the feature dimension. In our case, this entails assembling the two-dimensional matrices, each representing a different frequency band, into a four-dimensional array with dimensions (*s*, *f*, *h*, *w*), where *s* is the number of samples, *f* is the number of frequency bands, *h* is the height, and *w* is the width. However, to maintain compatibility with common deep learning frameworks, we can alternatively consider the last dimension *f* as the number of channels *c* in the feature maps, yielding a three-dimensional array of shape (*s*, *c*, *h*, *w*), where *c* corresponds to the number of channels. This process not only preserves the temporal and spectral information inherent in the original one-dimensional EEG signals but also enhances their representational power by introducing a spatial dimension. Consequently, it provides a richer set of features as input to subsequent deep learning models, facilitating improved performance in emotion recognition and other related tasks.

#### 3.4.2. Long-Short Time Memory Node 

EEG is a kind of time series data, and its characteristics have dynamic changes in time. Through LSTM, we can model the dynamic characteristics of EEG signals at different time points [26,27]. There may be long-term dependencies in EEG signals, that is, past emotional states may affect future emotional expressions. Traditional recurrent neural networks have the problem of gradient disappearance or gradient explosion, and LSTM effectively solves this problem by introducing gating mechanisms, especially forget gates and input gates. This enables LSTM to better capture long-term dependencies in sequence data, thereby improving the emotion recognition model’s ability to understand sequence data.

The structure of the LSTM node is shown in Figure 3, which reveals its internal working mechanism. In this design, *c_t_* and *h_t_* represent the memory state and hidden layer state of the unit respectively, which are key components of the model when processing sequence data. At the same time, *x_t_* and *y_t_* serve as the input and output of the model respectively, interacting with the external world. The *σ* symbol represents the sigma activation function, which plays an important role in LSTM. In addition, LSTM is equipped with three gating mechanisms to finely manage the flow of information:

The forget threshold, controlled by *f_t_*, determines which memory states should be discarded, thereby helping the model “forget” unimportant information.

The input threshold, controlled by *i_t_*, is responsible for screening which parts of $\tilde{c_{t}}$ should be included in the current memory state to achieve the effective integration of new information.

The output threshold, controlled by *o_t_*, determines which memory states should be read and used as output in the current time step, affecting the update of the hidden state *h_t_*.

As the memory state *c*_*t*−1_ is passed through the network, it first filters and discards information through the forget threshold, and then selectively absorbs new memory content through the input threshold. This mechanism ensures that the memory state of the model can be dynamically updated and optimized at each time step. Finally, after a series of operations, the content of the memory state is copied and passed to the function for processing, and then filtered by the output threshold, and finally a new hidden state *h_t_* is generated to prepare for the processing of the next time step.

#### 3.4.3. Bi-Convolutional-LSTM Layer 

We introduce a lightweight neurodynamic model to recognize complex emotional data from human. The convolution kernel of two-dimensional Convolutional layer operates in the temporal or spatial dimension, while the three-dimensional convolution kernel performs convolution in two dimensions and extracts useful classification information. CNN extracts local features of the input data by using convolutional layers and pooling layers [28,29]. The same convolution kernel is slid across the entire input data to detect similar features at different locations. 

In one-way LSTM, the hidden state at the current moment can only rely on the past input sequence, while bidirectional LSTM can use both past and future information to update the hidden state at the current moment. This helps reduce the risk of information loss, especially when processing longer sequences, and better preserves important information in the sequence. Bi-Convolutional-LSTM (BCL) maps the LSTM output to the space of attention weights through a fully connected layer. This fully-connected layer maps the hidden state of each time step output by the LSTM to a scalar value representing the importance of that time step.

#### 3.4.4. Dynamic Attention Mechanism

In tasks such as EEG emotion recognition, sequence data usually have a long-time span, in which some time steps may contain more critical information, while other time steps may contain relatively less or less important information. By introducing an attention mechanism [30], the model can dynamically adjust the degree of attention paid to different time steps, thereby more effectively capturing important features in sequence data, and giving different importance to information at different time steps.

Then, these scalar values are converted into attention weights via the softmax function such that the sum of all weights equals 1. These weights represent the relative importance of each time step, which means the degree to which the model should focus at that time step. The LSTM outputs are weighted and summed using the calculated attention weights to obtain a weighted representation. Specifically, for each time step, the LSTM output is multiplied with the corresponding attention weight, and then all weighted results are summed. This results in a weighted representation, where the contribution of each time step is determined by the attention weight. (7) is the formula of the attention mechanism:(7)$$Attention(Q,K,V)=Softmax(\frac{QK^{T}}{\sqrt{d_{k}}})$$

*Q, K, V* refer to Query, Key and Value respectively [31]. Query is obtained by linear transformation of the LSTM output and represents the hidden state of the current time step. Key represents the hidden state of the input sequence. They are used to measure the correlation between the hidden state of the current time step and the hidden states of other time steps. Value is the original representation of the LSTM output, that is, the hidden state without any linear transformation. It is used to represent the hidden state of each time step in the input sequence, and will be weighted and summed according to the calculated attention weight to obtain the final output. The weighted depiction is propagated through a fully connected layer to acquire the ultimate output forecast.

## 4. Discussion

### 4.1. Dataset Introduction

DEAP [32,33] is a physiological signal database used for emotion analysis. It contains physiological data and emotional ratings of multiple participants while watching multiple music video clips. Participants rated their emotional experience based on a discrete 9-point rating after watching each music video clip. These emotions include joy, excitement, anger, pressure, sadness, fear, surprise, calmness, and boredom. There are four values in each label: arousal, valence, dominance, and like. Valence represents the positive or negative nature of emotions. Arousal stands for the intensity or level of emotional activity. Dominance can reflect whether an individual is capable to control their emotional response, which is often expressed as confidence or a sense of authority. By dividing emotions into these three dimensions, they can help people better identify and classify emotions, and provide more accurate methods for emotion recognition.

To comprehensively evaluate students’ learning effectiveness, we propose an assessment framework based on the VAD (Valence-Arousal-Dominance) model, with a scoring range of 0–9 for each dimension. Specifically, Valence measures students’ interest and engagement in the learning content. Arousal reflects their mental activation and capacity for innovation. Dominance assesses their self-regulatory skills. Arousal, which directly relates to students’ attention and emotional activation, is assigned the highest weight (0.4) due to its importance in efficient learning. Valence and Dominance, reflecting students’ emotional disposition towards the lesson and their sense of control over the task, respectively, are each given equal weight (0.3) to balance their contributions to the learning effectiveness assessment. By assigning appropriate weights to each dimension and calculating score, we obtain a quantitative indicator that validly represents students’ learning effectiveness. This comprehensive approach not only facilitates a thorough understanding of students’ learning effectiveness, but also is helpful to develop personalized teaching plans and interventional strategies. The formula for calculating learning effectiveness score is shown in (8):(8)Score=0.3∗V+0.4∗A+0.3∗D

To illustrate the purpose of the formula, we have designed a classification standard based on scores derived from the VAD dimensions. The standard is shown in Table 2. This standard describes different learning states corresponding to different score ranges, each with specific characteristics, aimed at providing teachers with targeted teaching strategies.

To assess the practical applicability of the model, a graduate student (aged 23, female) and an undergraduate student (aged 19, female) were recruited as the experimental subjects in this study. Both subjects have signed the experimental consent form. As is shown in Figure 4, they wore non-invasive dry electrode caps and used OpenBCI Cyton to accurately capture raw EEG signals. In order to simulate real classroom settings and situations, the student was prompted to simulate a series of classroom states during the lessons. The EEG data acquisition was conducted at a high sampling rate of 256 Hz across eight channels, ensuring comprehensive and accurate data collection for subsequent analysis.

The SEED-IV [34] (SJTU Emotion EEG Dataset for Emotion Recognition with Four Emotions) dataset, developed by the Brain and Computational Intelligence Lab (BCMI) at Shanghai Jiao Tong University, is a comprehensive dataset for emotion recognition encompassing four emotion categories: happiness, sadness, fear, and neutrality. This dataset meticulously selected 72 movie clips as stimuli to induce these four emotions and recorded EEG and eye movement data from 15 healthy Chinese subjects. Each subject participated in three experiments, with at least a three-day interval between each session to mitigate any lingering emotional effects. The signals were acquired using a 62-channel ESI NeuroScan system with a sampling rate of 1000 Hz, which was subsequently downsampled to 200 Hz. These emotion categories are pre-defined and mutually exclusive, conforming to the characteristics of discrete data. In the dataset, each movie clip was designed to elicit one of these four discrete emotions, and participants were asked to self-evaluate based on these discrete emotion categories after watching.

### 4.2. Experimental Setup

#### 4.2.1. Labels Generation

In this paper, we employ the MATLAB interface to retrieve DEAP datasets that encompass emotional labels. Subsequently, we implement a binarization process for these labels, setting a threshold value of 5. Labels exceeding this threshold are designated as representing positive emotions, whereas those falling below are classified as negative emotions. This binarization step aims to simplify subsequent analyses. To align with the multiple temporal windows obtained from the differential entropy decomposition of EEG signals, each original emotional label is replicated and extended to match the corresponding number of windows. This ensures that each window is associated with its respective emotional label, facilitating the investigation of the relationship between EEG signal features and emotional states through deep learning.

The SEED-IV dataset has been down sampled to 200 Hz. We employ a specially designed bandpass filter to process the data, aiming to minimize artifacts to the greatest extent possible. Following this, we take into account the temporal dynamics of emotional states and employ the moving average method to smooth out the filtered data, thereby further eliminating irrelevant components. To ensure experimental balance, we still select differential entropy as the feature. The EEG signals from each experiment are segmented into individual samples, with a 1-s time window between each sample and no overlap. In the SEED-IV dataset, emotions are categorized into four distinct classes, hence the labels are mapped in a one-hot encoding format.

#### 4.2.2. Framework Settings

In this study, all experimental procedures were conducted within the Windows 11 environment. The construction and training of deep learning models were facilitated by PyTorch 2.1.0. All computational tasks were executed on NVIDIA GeForce 4070Ti GPU (NVIDIA Corporation, Santa Clara, CA, USA). BrainFlow library utilizes dual threading and dual process programming methods to achieve real-time raw data reading, which is used to obtain raw EEG signals on OpenBCI Cyton and analyze learning effectiveness.

A comprehensive overview of the crucial hyperparameters for each component is shown in Table 3.

The DEAP dataset employed in this experimental study originates from pre-processed dataset. We divided this dataset into three different subsets using a ratio of 6:2:2. The first 60% of the data is designated as the training set for training the model and adjusting its parameters. The remaining 20% is used as a validation set, allowing us to evaluate the performance of the model and make any necessary adjustments during the training process. The remaining 20% constitutes the test set for evaluating the final performance.

For the SEED-IV dataset, our experimental design adopted widely accepted standards proposed by Zhong et al. [9], utilizing the first 16 trials as the training set and the latter 8 trials as the test set. For each test sample, the model outputs a predicted category and then compares it with the true category to calculate accuracy. After completing the prediction of all test samples, we calculated the average accuracy of all these samples, in order to calculate the overall average accuracy.

For the optimization of our model during the training process, we have chosen the Adam optimizer. Adam [35] is a widely-used optimization algorithm that stands out due to its adaptive learning rates. In our experiment, we initiated the model with a learning rate of 0.0001, a batch size of 32, and conducted training for a total of 500 iterations to ensure thorough convergence and optimization of the model.

#### 4.2.3. Experimental Metrics

When evaluating the performance of emotion recognition models, it is common practice to adopt a multifaceted approach by selecting multiple evaluation metrics. This ensures a thorough assessment of the model’s capabilities. In the context of our experiment, we carefully chose three evaluation metrics: the F1 Score, Standard Deviation (STD), and Area Under the Curve (AUC). By combining these evaluation metrics, we aim to gain a holistic understanding of our emotion recognition model’s performance, ensuring that it not only achieves high accuracy but also demonstrates consistency and reliability in its predictions.
F1 Score


In classification tasks, F1 score [36] plays a key role as a comprehensive measure for evaluating model performance. By integrating two crucial indicators, precision and recall, the F1 score is designed to provide a more comprehensive and balanced perspective on assessment, particularly in situations involving imbalanced datasets. Precision predicts the ratio of positive instances to true positive instances, while the recall represents the proportion of true positive instances accurately predicted by the model as positive. The formula for calculating F1 score is shown in (9):(9)$$F1=\frac{2*Precision*Recall}{Precision+Recall}$$

As is shown in the (9), the F1 score spans from 0 to 1, where a larger number represents improved model effectiveness.
2.STD


In emotion recognition models, the role of standard deviation [37] is mainly reflected in the degree of data dispersion or variation, which is helpful to evaluate the performance stability of the model in different samples or batches of data. The formula is shown in (10):(10)$$S=\sqrt{\frac{\sum (x_{i}-\bar{x}{)}^{2}}{N-1}}$$
3.AUC


The AUC [38] serves as an indicator measuring a model’s capacity to prioritize positive instances over negative ones. Specifically, there are two approaches to determining the AUC: the first involves graphing the Receiver Operating Characteristic (ROC) curve and computing the enclosed area; the second entails arranging the anticipated scores of both positive and negative instances and determining the likelihood that a positive instance will be placed above a negative one. In the empirical studies presented here, we opt for the latter technique to ascertain the AUC figure, thereby assessing the efficacy of various classification models.

### 4.3. Experimental Results

According to Table 4, 3D-BCLAM achieved good results on the DEAP dataset. On the DEAP dataset, 3D-BCLAM s‘ valence classification accuracy, arousal classification accuracy and dominance classification accuracy were 95.47%, 95.83% and 96.88% respectively. It can be observed that different stimuli in different subjects tend to trigger different emotional dimensions.

Table 5 shows the typical experimental metrics of 3D-BCLAM in three states, which has good performance. To further confirm the impact of time series length in emotion recognition. 3D-BCLAM was tested on different time series lengths. Figure 5 shows that the classification accuracy of 3D-BCLAM differs in different time series, and the best performance was achieved when the sequence length is 8. Therefore, personalized models based on important features such as representative channels and appropriate time periods are of great value to achieve human emotion recognition.

As is shown in Table 6 and Table 7, 3D-BCLAM also achieved good results on the SEED-IV dataset. Among the 15 subjects, there were sessions where the accuracy rate surpassed 95%, indicating that our proposed model can be applied across different sessions with good generalization performance. However, there were also sessions where the accuracy rate fell below 65%, which might be caused by factors such as the quality of the collected EEG signals or similar brain activities among the subjects.

As shown in the Figure 6, the student’s learning effectiveness exhibited variation over a single testing cycle. During the initial time, the student’s learning scores remained above 5, indicating a favorable learning outcome. This can be attributed to the initial novelty of the course, which likely enhanced the student’s attention and engagement. However, in the subsequent test points, the student’s learning scores declined, gradually falling below 5, reflecting a moderate learning performance. This decline may be attributed to a gradual waning of the student’s patience and the emergence of a weariness towards learning as the course progressed. 

Besides, the disparity in learning outcome assessments between graduate and undergraduate students can be largely attributed to the distinctiveness in their respective educational orientations and cognitive paradigms. Graduate subject education emphasizes the cultivation of academic abilities, thereby fostering highly autonomous learning methods. This self-regulated learning mode not only deepens their subject knowledge, but also effectively improves their attention management skills through continuous exploration and practice. On the contrary, undergraduate education often emphasizes broad-based general education, and students typically receive more structured knowledge transfer. Although equally valuable, this model may lead to relatively inadequate self-regulation and attention control among undergraduate students compared to graduate students. In our research, we invited three experts in the field of education to conduct manual evaluations. The evaluations provided by these experts are consistent with the results generated by our proposed evaluation model, thus validating our research findings.

### 4.4. Baseline Comparison

A comparative evaluation of 3D-BCLAM was conducted alongside several other methodologies, including the Linear Support Vector Machine (Linear-SVM) [39], the Spiking Neural Network enhanced with Transfer Learning (SNN + TL) [40], the Attention-based CNN-RNN (ACRNN) [41], the Synchronous Brain Network (SBN-STM) [42], the SNN integrated with Infinite Impulse Response Filters (SNN + IIR) [43], the Fractal SNN (Fra-SNN) [44], the Functional Connectivity Network (FCN) [45], and the NeuCube SNN [46]. Given the similarities between 3D-BCLAM and these methods in terms of model architecture and feature extraction, they were selected as benchmarks for comparison. 

In addition, recently popular models based on graph neural networks and transformers were used for comparison, including Regularized Graph Neural Network (RGNN) [9], Attention-based Temporal Learner with Dynamic Graph Neural Network (AT-DGNN) [15], Dynamic graph convolutional neural networks (DGCNN) [47] and Emotion Recognition Transformer Net (ERTNet) [10]. The outcomes are presented in Table 8.

In comparison to these methodologies on DEAP, the accuracy variance ranged from 1.75% to 41.97%. Notably, when compared to approaches utilizing the frequency domain feature, Power Spectrum Density (PSD) (such as Linear-SVM and SNN + TL), 3D-BCLAM demonstrated enhancements of at least 12.72% and 11.61% in valence and arousal, respectively. Furthermore, when juxtaposed against methods employing DE (Fra-SNN), 3D-BCLAM exhibited improvements of 25.63%, 26.22%, and 23.68% in valence, arousal, and dominance, respectively. In comparison to these methodologies on SEED-IV, the accuracy variance ranged from 1.15% to 10.64%.

### 4.5. Ablation Study

To validate the efficacy of our proposed framework, we undertook a series of ablation studies [48]. Specifically, we systematically excluded certain components from the 3D-BCLAM model—namely, those modules instrumental in extracting EEG temporal features and those focusing on emotional attributes—and performed experiments. Ensuring consistency, the parameter inputs for each modified submodel were maintained akin to the original 3D-BCLAM configuration, thereby allowing us to rigorously assess the contributions of these components to the overall model performance. The results are shown in the Table 9.

## 5. Conclusions

In this study, we present 3D-BCLAM, for student learning effectiveness assessment through emotion recognition. The model integrates a Bi-Convolutional-LSTM architecture with a dynamic Attention Mechanism, utilizing differential entropy as the key emotional feature. A Bessel filter, combined with a Hamming window, is employed to preprocess data, transforming one-dimensional signals into three-dimensional representations to capture spatiotemporal relationships. This design enables more effective learning of data characteristics, improving the model’s performance in emotion recognition. Evaluated on the public dataset, 3D-BCLAM outperforms CNN, SVM, GNN, Transformer and hybrid SNN structures in recognizing emotions from EEG signals. By monitoring students’ emotional states, educators can gain valuable insights into their engagement, cognitive load, and overall learning experience. Despite its strengths, the model’s feature extraction capabilities could be further enhanced, particularly when handling complex EEG signals. Future work will involve applying the model in real-time EEG monitoring systems in educational environments, refining its performance on noisy or irregular data, and expanding its applicability across diverse student populations. These improvements would enhance the model’s robustness and provide deeper insights into the role of emotions in learning, potentially transforming educational practices.

## Figures and Tables

**Figure 1 sensors-24-07856-f001:**
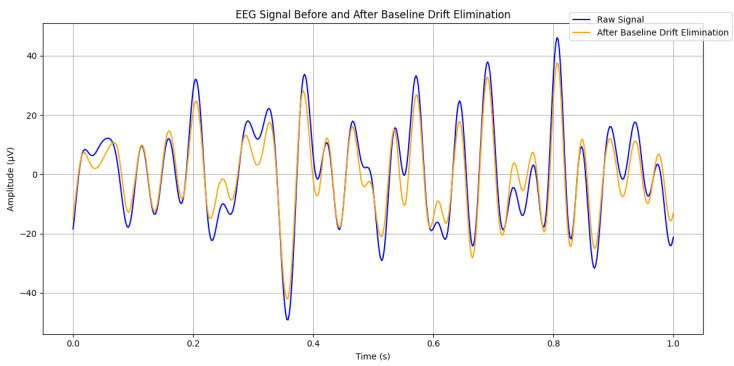
Baseline drift elimination based on wavelet transform.

**Figure 2 sensors-24-07856-f002:**
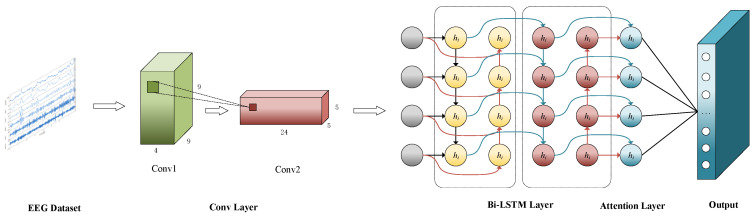
3D-BCLAM Framework.

**Figure 3 sensors-24-07856-f003:**
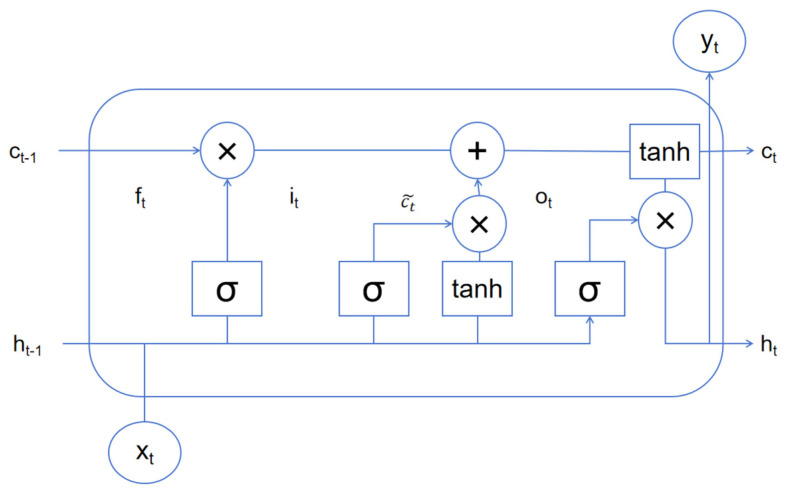
LSTM node.

**Figure 4 sensors-24-07856-f004:**
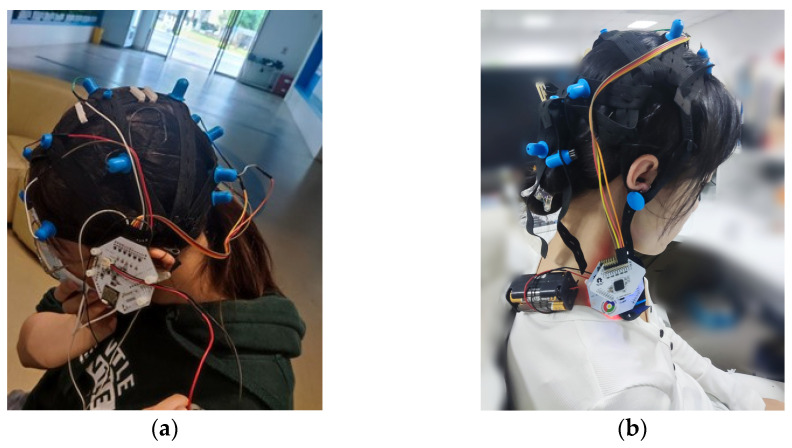
Practical application subjects. (**a**) a graduate student (aged 23, female); (**b**) an undergraduate student (aged 19, female).

**Figure 5 sensors-24-07856-f005:**
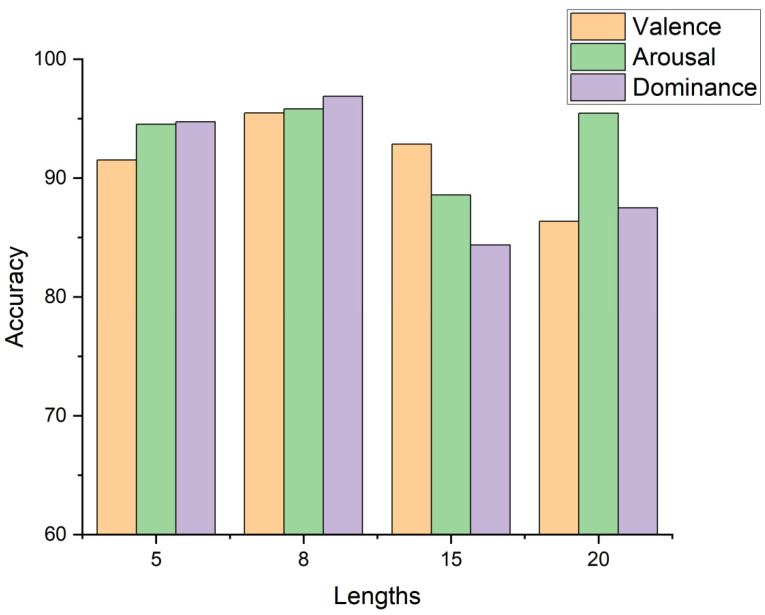
Applied time lengths of sequence and accuracy on DEAP.

**Figure 6 sensors-24-07856-f006:**
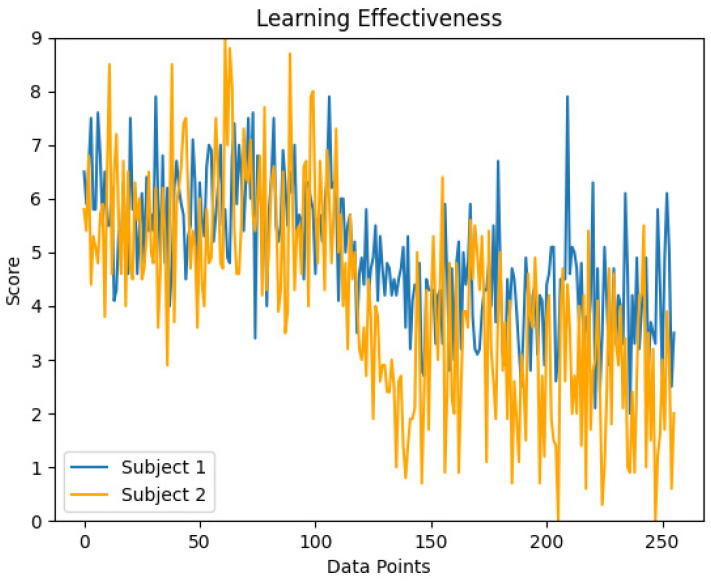
Learning Effectiveness.

**Table 1 sensors-24-07856-t001:** Various Bands and Characteristics of EEG Signals.

Type		Frequency	Characteristics
Theta		4–8 Hz	Generated in a latent state of consciousness
Alpha	Low Alpha	8–9 Hz	Blurred and confused before going to bed
	Mid Alpha	9–12 Hz	Relaxed body and mind, focused attention
	Fast Alpha	12–14 Hz	Highly alert
Beta	Low Beta	14–16 Hz	Relax but concentrate
	Mid Beta	16.5–20 Hz	Think and process receiving external information
	Fast Beta	20.5–31 Hz	Exciting or anxiety
Gamma		31–45 Hz	Stress relief and meditation

**Table 2 sensors-24-07856-t002:** Learning State Evaluation Standard.

Aggregated Score Range	Learning State	Characteristics
0–3	Ineffective	Lack of interest, distracted attention, weak self-management
3–4	Moderate	Moderate interest, but attention and self-control lacking
4–6	Good	Equitable learning state
6–9	Excellent	High engagement, focused attention, strong self-control

**Table 3 sensors-24-07856-t003:** Function Block and Hyperparameters of 3D-BCLAM.

Function Block		Hyperparameter
2D-Convolutional-layer	2D_Conv1	In_channels: 4 Out_channels: 24 Kernel size = 5 * 5
	2D_Conv2	In_channels: 24 Out_channels: 128 Kernel size = 2 * 2
Bi-LSTM layer		Batch_size = 32 Embedding_size: 16 bidirectional = True
Full-Connected layers	FC1 layer	Input_features = 128 * 2 * 2 Output_features = 96
	FC2 layer	Input_features = 96 Output_features = 16
Attention Mechanism layer		Input_features = 16 Output_features = Softmax

**Table 4 sensors-24-07856-t004:** 3D-BCLAM Performance on DEAP.

Subjects	VA ^1^	AA ^2^	DA ^3^
S1	92.52	96.62	95.15
S2	97.98	97.22	93.84
S3	94.06	100	100
S4	94.14	93.80	96.76
S5	94.38	95.21	100
S6	94.66	98.71	93.73
S7	95.26	94.37	92.67
S8	88.87	99.15	95.54
S9	95.78	94.90	97.66
S10	98.51	90.53	97.88
S11	93.30	97.01	98.44
S12	92.71	99.56	95.52
S13	98.26	95.66	100
S14	95.69	95.01	100
S15	96.18	95.99	94.03
S16	90.61	92.59	100
S17	100	94.41	96.84
S18	96.34	96.55	95.79
S19	92.92	97.35	100
S20	95.74	94.27	98.59
S21	95.81	98.11	96.72
S22	98.68	98.62	90.03
S23	94.52	96.89	98.61
S24	94.45	95.24	93.14
S25	97.86	95.75	97.96
S26	95.54	93.01	95.61
S27	90.37	92.02	97.72
S28	100	95.37	97.77
S29	88.8	97.81	96.31
S30	96.4	91.41	96.77
S31	98.03	100	96.73
S32	100	99.8	97.11

^1^ Valence accuracy. ^2^ Arousal accuracy. ^3^ Dominance accuracy.

**Table 5 sensors-24-07856-t005:** Evaluation Metrics of 3D-BCLAM on DEAP.

Metrics	Valence	Arousal	Dominance
Average Accuracy	95.47	95.83	96.88
Average STD	3.64	2.83	2.31
F1 Score	0.9555	0.9639	0.9718
AUC	0.9566	0.9642	0.9825

**Table 6 sensors-24-07856-t006:** 3D-BCLAM Performance on SEED-IV.

Subjects	Session 1	Session 2	Session 3
S1	93.96	96.34	82.39
S2	91.68	85.86	69.92
S3	63.01	78.04	71.39
S4	75.19	66.34	99.95
S5	97.69	75.25	93.08
S6	66.49	88.74	75.41
S7	69.40	91.16	61.43
S8	95.91	69.85	87.99
S9	85.82	85.24	62.48
S10	84.78	97.67	70.65
S11	65.20	90.26	74.34
S12	80.42	98.84	67.79
S13	80.59	68.13	90.84
S14	83.29	86.75	81.39
S15	65.79	88.86	67.87

**Table 7 sensors-24-07856-t007:** Evaluation Metrics of 3D-BCLAM on SEED-IV.

Metrics	SEED-IV
Average accuracy	80.52
Average STD	5.39
F1 Score	0.8148

**Table 8 sensors-24-07856-t008:** Comparison Methods.

Methods	VA ^1^	AA ^2^	DA ^3^	SA ^4^
Linear-SVM [39]	66.47	60.33	-	-
SNN + TL [40]	82.75	84.22	-	-
ACRNN [41]	93.72	93.38	-	-
SBN-STM [42]	78.00	78.30	-	-
ERTNet [10]	73.31	80.99	-	-
AT-DGNN [15]	83.74	86.01	-	-
SNN + IIR [43]	61.15	53.86	67.50	-
Fra-SNN [44]	69.84	69.61	73.20	-
FCN [45]	45.55	62.73	59.60	-
NeuCube SNN [46]	78.00	74.00	80.00	-
RGNN [9]	-	-	-	79.37
DGCNN [47]	-	-	-	69.88
3D-BCLAM	95.47 ± 3.64	95.83 ± 2.83	96.88 ± 2.31	80.52 ± 5.39

^1^ Valence accuracy. ^2^ Arousal accuracy. ^3^ Dominance accuracy. ^4^ SEED-IV accuracy.

**Table 9 sensors-24-07856-t009:** Ablation Study Results.

Methods	VA ^1^	AA ^2^	DA ^3^	SA ^4^
3D-BCLAM without AM	90.88 ± 2.97	90.71 ± 4.66	88.41 ± 3.85	76.45 ± 8.31
3D-BCLAM without Bi-LSTM	66.25 ± 5.31	62.93 ± 5.18	64.16 ± 4.95	49.67 ± 5.87
3D-BCLAM without Bi-LSTM and AM	64.47 ± 7.13	60.31 ± 6.48	62.13 ± 5.22	48.35 ± 6.53
3D-BCLAM	95.47 ± 3.64	95.83 ± 2.83	96.88 ± 2.31	80.52 ± 5.39

^1^ Valence accuracy. ^2^ Arousal accuracy. ^3^ Dominance accuracy. ^4^ SEED-IV accuracy.

## Data Availability

Data are contained within the article.

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
