# Peer review of "3D-BCLAM: A Lightweight Neurodynamic Model for Assessing Student Learning Effectiveness"

_sensors, 2024, doi:10.3390/s24237856_

Round 1
Reviewer 1 Report
Comments and Suggestions for Authors
The paper introduces an innovative lightweight neuro-dynamic model, 3DBCLAM, which combines Bidirectional Convolutional Long Short-Term Memory with a dynamic Attention Mechanism to effectively capture temporal emotional dynamics. This model, utilizing emotion classification, provides a comprehensive assessment of student learning effectiveness across both cognitive and affective dimensions. While the topic and methodology demonstrate innovation, there are several areas within the paper that require further attention and clarification:
(1) The validation of the model is solely conducted on the DEAP dataset. It is recommended that the model's effectiveness be tested on other classic datasets to substantiate its validity. Additionally, reviewing recent literature to compare the performance of current leading models on the DEAP dataset would be beneficial.
(2) The paper proposes a learning effectiveness score using different weights for the V, A, D indices to synthesize a score, yet the rationale behind this weighting is not explained.
(3) The introduction and related work sections of the paper lack sufficient references. A more thorough investigation is suggested to discuss the current state of research, including the use of Graph Neural Network (GNN) models and Transformer-based models in EEG-based emotion recognition.
(4) Regarding the comparative algorithms, it is necessary to compare with recent models, especially those related to the aforementioned GNN and Transformer models.
Overall, the paper holds potential, and I recommend revisions based on these comments.
Comments on the Quality of English LanguageThe authors have presented their research clearly and coherently, making it easy to understand the concepts and methodologies discussed.
Author Response
Responses to Reviewer #1
Comment 1: The validation of the model is solely conducted on the DEAP dataset. It is recommended that the model's effectiveness be tested on other classic datasets to substantiate its validity. Additionally, reviewing recent literature to compare the performance of current leading models on the DEAP dataset would be beneficial.
Response: We sincerely appreciate your valuable comments and suggestions, particularly your concern regarding our model's validation being solely based on the DEAP dataset. We fully acknowledge the critical importance of testing our model's effectiveness on other classic datasets to further substantiate its validity and generalization capability.
In response to your recommendation, we have supplemented our evaluation by including the SEED-IV dataset to assess our model's performance. The SEED-IV (SJTU Emotion EEG Dataset for Emotion Recognition with Four Emotions) dataset, developed by the Brain and Computational Intelligence Lab (BCMI) at Shanghai Jiao Tong University, is a comprehensive dataset for emotion recognition encompassing four emotion categories: happiness, sadness, fear, and neutrality. This dataset meticulously selected 72 movie clips as stimuli to induce these four emotions and recorded EEG and eye movement data from 15 healthy Chinese subjects. Each subject participated in three experiments, with at least a three-day interval between each session to mitigate any lingering emotional effects. The signals were acquired using a 62-channel ESI NeuroScan system with a sampling rate of 1000Hz, which was subsequently downsampled to 200Hz. By conducting experiments on the SEED-IV dataset, we aim to evaluate our model's performance in diverse contexts, ensuring that our findings are not confined solely to the DEAP dataset. This expansion will undoubtedly contribute to a more robust and generalizable model, enhancing its applicability in real-world scenarios.
Furthermore, we have conducted a thorough review of the recent literature and compared their performance on the DEAP dataset with our proposed model. This comparison will not only provide insights into the strengths and weaknesses of our model but also contribute to the broader field of emotion recognition by highlighting key trends and advancements.
Comment 2: The paper proposes a learning effectiveness score using different weights for the V, A, D indices to synthesize a score, yet the rationale behind this weighting is not explained.
Response: We sincerely appreciate your valuable feedback on our paper. You have rightly pointed out that the rationale behind the weighting of the V (Valence), A (Arousal), and D (Dominance) indices in our proposed learning effectiveness score was not fully explained. In setting these weights, we took into careful consideration the distinct roles each index plays in the learning process. Arousal, which directly relates to learners' attention and emotional activation, was assigned the highest weight (0.4) due to its pivotal role in efficient learning. Valence and Dominance, reflecting learners' emotional disposition towards the content and their sense of control over the task, respectively, were each given equal weight (0.3) to balance their contributions to the learning outcome assessment, albeit recognizing their somewhat less direct impact compared to Arousal. We acknowledge the importance of transparency and justification in weight assignment for research reproducibility and credibility. In future studies, we will further explore and validate these weights to more accurately capture the influence of emotional factors on learning effectiveness. We welcome further input and suggestions from the research community to advance this field of study.
Comment 3: The introduction and related work sections of the paper lack sufficient references. A more thorough investigation is suggested to discuss the current state of research, including the use of Graph Neural Network (GNN) models and Transformer-based models in EEG-based emotion recognition.
Response: Thank you very much for your meticulous review and invaluable feedback on our paper. We fully acknowledge and appreciate your point regarding the need for more references in the introduction and related work sections, as well as a more thorough discussion of the current research status, particularly concerning the application of Graph Neural Network (GNN) models and Transformer-based models in EEG-based emotion recognition. In response, we have conducted an extensive literature review to gather recent studies in the field of EEG emotion recognition that involve GNNs and Transformer models, ensuring that our introduction and related work sections comprehensively reflect the current research trends and latest advancements. We have highlighted the added content in yellow.
Comment 4: Regarding the comparative algorithms, it is necessary to compare with recent models, especially those related to the aforementioned GNN and Transformer models.
Response: Thank you very much for your thorough review and valuable suggestions. We fully agree that comparing our algorithm with the latest models, particularly those related to Graph Neural Networks (GNNs) and Transformer models, is crucial for evaluating its performance. In response to your recommendation, we selected the RGNN, DGCNN and AT-DGNN models, based on graph neural network architecture, for comparison. These models primarily capture local and global relationships between different EEG channels across various brain regions. Experimental results demonstrate that we achieved superior performance on both the DEAP and SEED-IV datasets, thereby validating the effectiveness of our proposed model.
Additionally, we have included comparisons with Transformer variants applicable to similar tasks, ensuring a fair experimental setup to accurately assess the strengths and limitations of different models. We selected the ERTNet model based on the transformer architecture as a comparison: they developed an interpretable end-to-end EEG emotion recognition framework based on a hybrid CNN and transformer architecture, which utilizes the transformer to process the feature map, integrate high-level spatiotemporal features, and thus recognize the current emotional state. The experimental results indicate that we have achieved better results on the DEAP dataset, which can prove the effectiveness of our proposed model. Furthermore, we have updated the introduction and related work to reflect the latest research developments and discuss the relationship between our work and the newly included comparison models. We deeply appreciate your suggestions, as they will significantly improve the quality and comprehensiveness of our work.

Reviewer 2 Report
Comments and Suggestions for Authors
The manuscript is devoted to the extremely important problem of improving the quality of education and undoubtedly deserves attention of specialists in the field of machine vision in education. I hope that after serious revision the manuscript will be published in this journal.
However, a number of comments do not allow me to give a positive answer at this point.
One of the key technologies that the authors used for data preprocessing is wavelets. However, neither EEG nor processed images are provided in the manuscript. I advise the authors to provide these images in their manuscript. Then the manuscript will be clearer and more understandable. This manuscript can be cited as an example: doi.org/10.1007/s11416-023-00500-2
Line 191: Clarification is required regarding the wavelet basis function.
The article is filled with a lot of information, knowledge of which is a must for any researcher in the field of machine vision:
Line 474-486: Adam is the most popular optimiser at the moment. Authors better make a link.
Line 490-512: Experimental metrics. Any student of classification methods knows them. The authors would be better off making a reference.
Line 542: It is unclear how the authors calculated average accuracy. This metric is very dependent on the sampling structure. The authors did not provide information about this structure.
It is unclear whether such research can be extended to a broad category of students. A postgraduate student is a person who has already developed attention management skills. I believe that the results obtained in the experiment can be considered close to optimal. But it would be interesting to see and compare them with the results of subjects who do not have the skills that a graduate student has.
To summarise. The authors should thoroughly revise their manuscript. As it stands, it is difficult to read and poorly illustrated. There are sections that could have been replaced by a single reference, while at the same time the main methods used by the authors are sparingly explained.
Author Response
Responses to Reviewer #2
Comment 1: One of the key technologies that the authors used for data preprocessing is wavelets. However, neither EEG nor processed images are provided in the manuscript. I advise the authors to provide these images in their manuscript. Then the manuscript will be clearer and more understandable. This manuscript can be cited as an example: doi.org/10.1007/s11416-023-00500-2
Response: Thank you for your comments, we acknowledge the importance of including EEG and processed images in the manuscript to enhance clarity and understanding. Following the example of the cited manuscript (doi.org/10.1007/s11416-023-00500-2), we have supplemented the baseline drift elimination results of the original EEG based on wavelet transform (taking a single channel as an example), in order to better illustrate the use of wavelets in our data preprocessing technique.
Comment 2: Clarification is required regarding the wavelet basis function. The article is filled with a lot of information, knowledge of which is a must for any researcher in the field of machine vision:
Response: Thank you for your feedback. We have added relevant content about wavelet basis functions in the manuscript. The wavelet basis function is a fundamental concept in signal processing, referring to a specific set of mathematical functions that enable the efficient decomposition of signals into different frequency and time scale components. These functions are characterized by their localization properties in both time and frequency domains, which allow for precise capture of signal features. When selecting a wavelet basis function for research purposes, it is crucial to consider factors such as signal characteristics, orthogonality, compact support, and vanishing moments to ensure the effectiveness and accuracy of the analysis. We appreciate your recognition of the importance of this knowledge in the field of signal processing.
Comment 3: Adam is the most popular optimiser at the moment. Authors better make a link.
Response: Thank you for the suggestion. We acknowledge the popularity and effectiveness of the Adam optimizer in various applications. We have added a reference to the Adam optimizer in the manuscript
Comment 4: Experimental metrics. Any student of classification methods knows them. The authors would be better off making a reference.
Response: We appreciate the reviewer's comment regarding the experimental metrics used in classification methods. To enhance the clarity and rigor of our work, we have included explicit references to standard evaluation metrics including F1-score, standard deviation and AUC as appropriate. This will not only provide a solid foundation for our experimental results but also facilitate a better understanding and comparison of our work with existing literature in the field.
Comment 5: It is unclear how the authors calculated average accuracy. This metric is very dependent on the sampling structure. The authors did not provide information about this structure.
Response: First and foremost, we would like to express our sincere gratitude for your meticulous review of our work and the invaluable feedback provided. Your questions regarding the calculation method of average accuracy and the sampling structure are indeed pertinent, prompting us to elaborate further on these aspects to enhance the clarity and credibility of our paper.
In the DEAP dataset, we initially segmented the EEG signals for each trial and each channel into two parts: pre-trial signals (the first 3 seconds) and trial signals, with a sampling frequency of 128Hz. The corresponding emotional labels were dichotomized based on the subjects' evaluation scores (scores greater than 5 were classified as positive, and scores less than or equal to 5 as negative) and converted into one-hot encoding. We then divided the one-dimensional data by frequency bands, transformed the data of each band into 9x9 two-dimensional matrices, and stacked these matrices into three-dimensional dataset samples. The processed dataset was subsequently split into 60% for training, 20% for validation, and 20% for testing. The sampling structure was based on sequential data, where each sample consisted of a sequence with multiple time steps, and each time step had multi-dimensional features. Accuracy was calculated as the number of correct predictions divided by the total number of samples. To obtain the final average accuracy, we computed the arithmetic mean of the accuracies across all subjects.
For the SEED-IV dataset, our experimental design adhered to widely accepted standards within the field, utilizing the first 16 trials as the training set and the latter 8 trials as the test set. For each test sample, the model outputted a predicted category, which we then compared to the true category to calculate accuracy. After completing predictions for all test samples, we computed the average accuracy across all these samples to obtain the overall average accuracy.
Comment 6: It is unclear whether such research can be extended to a broad category of students. A postgraduate student is a person who has already developed attention management skills. I believe that the results obtained in the experiment can be considered close to optimal. But it would be interesting to see and compare them with the results of subjects who do not have the skills that a graduate student has.
Response: We are deeply grateful for your invaluable feedback on our research, particularly regarding the breadth and representativeness of the study population. We fully comprehend and concur with your perspective, and in response, we plan to include an undergraduate student as a new participant to initially investigate variations in attention management skills across different academic stages. Although primary and secondary school students are currently unable to participate directly due to age, cognitive, and ethical constraints, they are considered potential subjects for future research. We pledge to continuously strive for rigor and comprehensiveness in our study, and we look forward to achieving even more fruitful results in our future endeavors.
Comment 7: To summarise. The authors should thoroughly revise their manuscript. As it stands, it is difficult to read and poorly illustrated. There are sections that could have been replaced by a single reference, while at the same time the main methods used by the authors are sparingly explained.
Response: We sincerely appreciate the reviewer's constructive feedback on our manuscript. We acknowledge that the current version may lack clarity and sufficient illustration, and we have thoroughly revised the text to enhance readability and comprehension. We are committed to addressing these issues and improving the overall quality of the manuscript.

Reviewer 3 Report
Comments and Suggestions for Authors
Idea of the paper: Good and innovative.
Abstract: Detected as 66% AI-generated text.
Conclusion: 100% AI-generated text.
Introduction: The beginning of the introduction is also detected as 100% AI-generated.
The other parts of the text were not checked.
In a scientific article, this is unacceptable.
Comments on the Quality of English Language
The English is straightforward but can be improved for conciseness and fluency.
Author Response
Responses to Reviewer #3
Comment 1: Idea of the paper: Good and innovative. Abstract: Detected as 66% AI-generated text. Conclusion: 100% AI-generated text. Introduction: The beginning of the introduction is also detected as 100% AI-generated. The other parts of the text were not checked. In a scientific article, this is unacceptable.
Response: Thank you very much for reviewing our paper and providing valuable feedback. We are deeply honored to have your recognition and believe that our paper ideas are innovative and valuable. However, regarding your concern that some of the text in the paper may have been generated by AI, we need to clarify solemnly: in the process of writing this paper, we did not use any AI tools to generate the text content. We understand that the application of AI technology in text generation is becoming increasingly widespread, and we also understand that this may lead to some misunderstandings. But please believe that we always adhere to the principles of academic integrity and originality, and all paper content is the result of in-depth research, discussion, and writing by our team members.
Based on your valuable feedback, we have revised the manuscript in terms of conciseness and fluency, and conducted a re check on the revised manuscript. The similarity rate and AI generated detection rate meet the requirements of the journal. We highly value your review comments and understand your strict requirements for academic integrity. Therefore, we kindly request that you review our paper again and consider our clarifications and explanations.

Round 2
Reviewer 2 Report
Comments and Suggestions for Authors
I thank the authors of the article for their comprehensive answers. My comments were taken into account in the new version of the article. In my opinion, the changes in this version of the article have significantly improved it. There are no further questions to the authors.
Author Response
Thank you very much for your affirmation and recognition! We are pleased to hear that you have provided valuable feedback on our manuscript, and these suggestions have been fully reflected in the new version of the manuscript. We are honored to have the opportunity to improve the manuscript based on your suggestions, and we are pleased that these changes have been recognized by you, significantly enhancing the quality of the manuscript. Looking forward to the opportunity to collaborate again in the future and jointly promote the progress of academic research. Thank you!
Reviewer 3 Report
Comments and Suggestions for Authors
The results obtained using other tools still indicate the use of AI in writing this article

Author Response
Responses to Reviewer #3
Comment 1: The results obtained using other tools still indicate the use of AI in writing this article.
Response: Dear reviewer, thank you very much for reviewing our article and providing valuable feedback. We solemnly declare that this article was entirely handwritten by our team members and did not use any AI writing tools. We understand your emphasis on academic integrity and provide full-text plagiarism reports, AI detection reports, partial raw EEG data collected from our own human experiments, and processed EEG data. Based on the issues you pointed out, we have made further modifications. We look forward to further communication with you to eliminate any misunderstandings, and thank you for your careful review.
